

# Soil water migration in the unsaturated zone of semi-arid region in China from isotope evidence

Yonggang Yang[1,2], Bojie Fu[2]

[1] Institute of Loess Plateau, Shanxi University, Taiyuan, Shanxi 030006, China;

[2] Research Center for Eco-Environmental Sciences, Chinese Academy of Sciences, Beijing, 100085, China

*Correspondence to:* Bojie Fu (bfu@rcees.ac.cn)

**Abstract.** Soil water is an important driving force of the ecosystems, especially in the semi-arid hilly and gully region of northwestern Loess Plateau in China. The mechanism of soil water migration in the reconstruction and restoration of Loess Plateau is a key scientific problem that must be solved. Isotopic tracers can provide valuable information associated with complex hydrological problems, which is difficult to obtain by other methods. In this study, the oxygen and hydrogen isotopes are used as tracers to investigate the migration processes of soil water in the unsaturated zone in the Loess Plateau of arid region in China. Samples of precipitation, soil water, plant xylems and plant roots are collected and analysed. The conservative elements D and $^{18}$O are used as tracers to identify variable source and mixing processes. The mixing model is used to quantify the contribution of each end member and calculate mixing amounts.The results show that the isotopic composition of precipitation in the Anjiagou River basin is affected by isotopic fractionation due to evaporation. The isotopic compositions of soil waters are plotted between or near the local meteoric water lines, indicating that soil waters are recharged by precipitation. The soil water migration is dominated by piston-type flow in the study area, but rarely preferential flow.Water migration exhibited a transformation pathway from precipitation to soil water to plant water. $\delta^{18}$O and $\delta$D are enrichment in the shallow (<20cm depth) soil water in most soil profiles due to evaporation. The isotopic composition of xylem water is close to that of soil water at the depth of 40-60 cm. These values reflect soil water signatures associated with *caraganakorshinshiikom* uptake at the depth of 40-60 cm. Soil water from the surface soil lay (20-40cm) contributed to 6%-12% of plant xylem water, while soil water at the depth of 40-60 cm is the largest component of plant xylem water (range from 60% to 66%), soil water below 60 cm depth contributed 8%-14% to plant xylem water, and only 5%-8% is derived directly from precipitation.This study investigates the migration process of soil water, and

identifies the source of plant water, and finally provides a scientific basis for identification of model structure and

parameter. It can provide a scientific basis for ecological water demand, ecological restoration, management of water

resources and the improvement of water benefit on each scale.

**Keywords**: soil water; xylem water; isotopes; migration; unsaturated zone

## 1 Introduction

Water in the soil environment plays a crucial role as a carrier of dissolved and solid species and as a reservoir in the

hydrological cycle. Soil water possesses a small proportion (only 0.05 %) within the hydrological cycle, but it is vital for an

ecosystem and affects spatial and temporal processes at different scales (Busari et al., 2013).Understanding soil water

migration in the unsaturated zone is essential to describe the movement of salt, carbon, nitrogen and other nutrient. Soil

water migration plays an important role in the processes of infiltration, evaporation, transpiration and percolation, hydraulic

conductivity and water uptake capacity of soils in the unsaturated zone. The traditional methods have been carried out to

study the movement of soil water such as hydrologic experiment, intensive observations, modeling and remote sensing (Luo

et al., 2013; Yang et al., 2013; Carucci et al., 2012; Buttle et al., 2000; Barnes et al., 1988). However, soil water migration is

a complex nonlinear and inhomogeneous flow process. Therefore, it is difficult to model on the basis of Darcy's law

exclusively, and is also difficult to obtain by other techniques. Climatic conditions, soil texture and soil structure, antecedent

moisture and vegetation cover exert influences on the movement of soil water (Kidron et al., 2013).

Isotopic tracers can provide valuable information on the complex hydrological problems, such as the runoff processes,

residence time, runoff pathway, the origin and contribution of each runoff component (Arny et al., 2013; Ohlanders et al.,

2013;Yang et al., 2012a; McInerney et al, 2011; Maurya et al., 2011; Kevin et al., 2010; Wu et al., 2009; Liu et al., 2008).

The isotopes of D and $^{18}$O are conservative and do not react with clay minerals and other soil materials. D and $^{18}$O are widely

used to investigate ecological and hydrological processes. Stable isotopes can provide information about mixing, transport

processes and residence time of water within a soil profile in the unsaturated zone.

Different flow mechanisms result in different isotopic profiles. Stable isotope compositions of soil water, plant xylems

and precipitation can be obtained to identify soil water migration processes such as infiltration, evaporation, transpiration



and percolation (Caley et al., 2013 ; Yang et al., 2012b; Catherne et al., 2010; Stumpp et al., 2009; Gazis and Feng, 2004).

Zimmerman et al. (1967) firstly applied stable isotopes to study the soil water profile, showing that evaporation at the surface of a saturated soil column causes deuterium enrichment near the surface that decreases exponentially with depth. Robertson and Gazis (2006) studied the seasonal trends of oxygen isotope composition in soil water fluxes at two sites along the climate gradient. The hydrogen and oxygen isotopes are used to study the transforming of precipitation, soil water and groundwater of typical vegetation in Taihang mountain area (Song et al., 2009). Gazis and Feng (2004) compared the

oxygen isotope compositions of precipitation and soil water from profiles at six sites with different soil textures. $\delta D$ and $\delta^{18}O$ in soil water has been observed in many studies, including column experiments using sand or soil columns and field observations of unsaturated soils (Singh et al.,2013; Návar et al.,2011; Gehrels et al., 1998;Allison and Hughes, 1983).

Stable isotope is the evidence for assessing plant water sources according to their variations caused by equilibrium and kinetic isotopic fractionation mechanisms (Haverd et al. 2011; Zhao et al. 2011;Bhatia et al., 2011). Several studies (Liu et

al.,2013; Mathieu et al., 1996; Allison et al., 1985) have shown that transpiration do not cause isotopic fractionation of soil water. The isotope $\delta D$ and $\delta^{18}O$ is a powerful tool to determine the water sources utilized by plants in the field, since $\delta D$ and $\delta^{18}O$ values of xylem water reflect those of the water sources utilized (Yang et al., 2011; Lu et al.,2011). The naturally occurring vertical gradients of $\delta D$ and $\delta^{18}O$ in soil water provide similar information about plant water uptake depth from soils. A number of studies have used isotopes to characterize soil water movement in one location and thus one climatic

regime.(Manzoni et al., 2013; Liu et al., 2011;Stumpp et al., 2009; Gazis and Feng, 2004).

Soil moisture is an important driving force of the ecosystems, especially in the northwestern Loess Plateau. Characterized by dry climate, less precipitation, more evaporation and thicker soil layers, groundwater in this region is difficult to use due to the depth of water table. Thus, soil water is almost the only water resource in the study area, which has become the only factor controlling agricultural production and ecological restoration. It is necessary to investigate the

mechanism of soil water migration in the loess plateau. Stable isotopic can provide valuable information on the mechanism of soil water migration. At present, a large number of research focuses on the soil water recharge by precipitation and its isotope variation characteristics. There is some research on the interaction of soil water, plant water and precipitation. These studies are mostly on the individual scale in one specific region (Heathman et al., 2012; William et al., 2010; Ferretti et al.,

2003). However, the research on the migration process of soil water, plant water, precipitation and groundwater in the

unsaturated zone based on the isotopic technique is still rarely. At present, the mechanism of soil water migration in the

Loess Plateau is the key scientific problem to be solved.

Therefore, this study investigates soil water migration processes in the unsaturated zone in the hilly and gully region of

Loess Plateau using isotopes, integrated with sampling in the field, experimental observation and laboratory analysis.

Samples include the soil water, plant xylem and root etc. The objectives of this research are: (1) to probe into the migration

process and variation of plant water, soil water, precipitation and groundwater in the unsaturated zone; and (2) to identify

each potential water source uptake by plant, and evaluate their contributions of each potential water source in the unsaturated

zone. It can provide a scientific basis for ecological water demand, ecological construction, management of water resources

and the improvement of water benefit on each scale.

## 2. Materials and methods

### 2.1 The experimental site

The study area is located in Anjiagou River basin in the Dingxi City of China, with latitudes 34°26′-35°35′N, and

longitudes 103°52′-104°39′E. The climate is semi-arid, with an average annual temperature of 6.3℃. The annual mean

precipitation is 420mm. The mean annual evaporation is 1510mm. The aridity index is 1.15. Precipitation is low and

unevenly distributed in temporal and spatial. This area is a part of the typical semi-arid in the hilly and gully region of

Loess Plateau, with the altitude ranging from 1900 m to 2250 m.

The watershed area is 8.91km$^2$, which belongs to the hilly and gully region of Loess Plateau. Gully density is

3.14km/km$^2$, and the ditch depth ranges from 30 to 50 m. Soil type is yellow loessal soil and saline soil, and the average

thickness ranges from 40 to 60 m. The soil density of soil layer ranges from 1.1 to 1.4 g/cm$^3$, average soil porosity is 55%.

Soil structure has a vertical joint, the nature of soil is loose and its wet collapsibility is serious. The grassland and shrubland

ecosystems are the most extensively dominant ecosystems in the Anjiugou river basin. As it is a representative area of Loess

Plateau area, the Anjiagou River basin is a suitable area for soil water study in semi-arid region.



## 2.2 Sample collection and field experiment

The research was carried out in Anjiagou River basin in the hilly and gully region of Loess Plateau from 2013 to 2015. Samples of precipitation, soil, xylem, stem, leaf and root were collected in the study area. The locations of the sampling sites are shown in Fig.1. Precipitation was collected after each rainfall event. Precipitation was filtered and transferred to sealed glass vials prior to analysis. Plant xylem, stem, leaf and root of *caraganakorshinshiikom* were collected at each sampling site, respectively. For sap water analysis, a plant twig (0.5 to 1cm in diameter and about 5 cm in length) was cut from a main branch or trunk, cortical and phloem were peeled off. Stem samples (about 0.5 to 2 cm diameter and 5 cm length) were obtained. All samples were kept in the 8 ml glass bottle, sealed with packaging tape to reduce evaporation. Five glass bottles were collected for plant in each sampling site. All samples were collected at least monthly. Altogether 27 samples sites and 396 samples were collected, and sealed with parafilm. The samples were taken back to the laboratory and preserved at 4 ℃, and samples of soil, plant xylem, plant stem and leaf were refrigerated at -20 ℃ until analysis.

Soil samples in the unsaturated zone were collected at 10 cm intervals for the first 40 cm, 20 cm intervals from 40 to 100 cm, 30 cm intervals from 100 to 130 cm. Maximum depths of sampling ranged up to 130 cm (Plant root is rarely found below 100 cm in the study area). At each sampling site, soil moisture (volumetric soil water content) was obtained with time domain reflectometry (TDR) in the field manually at 0-10, 10-20, 20-30, 30-40, 40-60, 60-80, 80-100and 100-130 cm. Soil moisture content was determined by oven drying method simultaneously.

## 2.3 Laboratory analyses

Water was extracted from soil, xylem, root, stem and leaf by cryogenic vacuum distillation method, and the extracted water (2 to 10 ml) was trapped at liquid–nitrogen temperature. Vacuum distillation was considered as a reliable and accepted method. Water samples were filtered through 0.2μm Millipore membrane for trace elements analyses. Isotopes $\delta^{18}O$ and $\delta D$ were measured in the Key Laboratory of Ecohydrology and River Basin Science of Cold and Arid Region Environmental and Engineering Institute, Chinese Academy of Sciences. $\delta^{18}O$ of samples was analyzed by a Euro-PyrOH elemental analyzer at a temperature of 1300°C, and $\delta D$ was analyzed at a temperature of 1030°C. At both temperatures the reaction products were analyzed on a GV ISOPRIMETM continuous flow IRMS. In order to eliminate the memory effect of online





and continues flow, every sample was analyzed 6 times. The analytical precision of δD determinations was ±1‰, and that of

$\delta^{18}$O±0.2‰. Isotopic concentration was expressed as δ-per million (‰) relative to the Vienna Standard Mean Ocean Water,

according to the follow Eq. :

$$\delta^{18}O(\text{‰}) = \frac{(^{18}O/^{16}O)_{sample} - (^{18}O/^{16}O)_{SMOW}}{(^{18}O/^{16}O)_{SMOW}} \times 1000, \qquad \delta D(\text{‰}) = \frac{(D/H)_{sample} - (D/H)_{SMOW}}{(D/H)_{SMOW}} \times 1000$$

## 2.4 Statistical analysis

When $n$ isotope is used to determine the proportional contributions of $i+1$ sources to a mixture, standard linear mixing

models can be used to mathematically solve for the unique combination of source proportions that conserves mass balance

for all n isotopes (Phillips et al., 2013). In this study, the conservative elements D and $^{18}$O are utilized to calculate mixing

amounts of potential water source. The mixing model is used to calculate the contributions of each respective end-member.

The mixing model is described on the basis of the mass balance Eq.:

$$\delta D_P = \sum_{i=1}^{n} f_i \delta D_i$$

$$\delta^{18}O_P = \sum_{i=1}^{n} f_i \delta^{18}O_i$$

$$I = \sum_{i=1}^{n} f_i$$

Where δD and $\delta^{18}$O are concentrations of the tracers; $I$ refers to potential water source; $\delta D_p$ and $\delta^{18}O_p$ refer to plant waters; $f_i$

is the fraction of each component contributing to plant water.

## 3. Result

### 3.1 Isotopic composition of precipitation

Yurtsever and Gat (1981) modified Craig's global water line, and made a more accurate global precipitation linear

relationship between δD and $\delta^{18}$O of δD=8.17$\delta^{18}$O+10.56, which is the Global Meteoric Water Line (GMWL). Figure 2

shows the relationship between the mean $\delta^{18}$O and mean δD in precipitation, which is defined as the local meteoric water

line (LMWL) with an Eq. of δD=7.41$\delta^{18}$O+7.22, R$^2$=0.95. The $\delta^{18}$O and δD compositions of the precipitation in the

Anjiagou River catchment are extremely variable, ranging from -118.36‰ to -22.32‰, and from -16.35‰ to -2.19‰,



respectively. There are significant differences in the composition of D and $^{18}$O under different precipitation event. The difference reflects the extreme nature of climate and the complexity of moisture sources in the arid region. Compared with

145 GMWL, the slope and intercept of LMWL are lower than that of GMWL (Fig.2), indicating that they are affected by the local climate and environment with less precipitation and lower humidity. When the slope is low, the implication is that precipitation is subject to evaporation. The Anjiagou River basin is located in the northwestern inland region, and it is difficult for the sea water vapor to reach directly. This indicates that the significant imbalance of isotope dynamic fractionation exists, which are strongly affected by strong solar radiation and evaporation in the local environment, leading

to the enrichment in $\delta^{18}$O and $\delta$D. Therefore, the slope of LMWL is low, and intercept of LMWL decreases with the slope.

**3.2 The variations of δD and δ18O in soil water of vertical profiles**

Stable isotope compositions of soil water are presented in Table1. The measured D and $^{18}$O of soil water range from -72.42‰ to -37.05‰, and -11.74‰ to -3.57‰, respectively. By comparing the isotopic composition of soil water with that of precipitation, the isotope compositions of soil water in the unsaturated zone are relatively enriched. Therefore, it could be

beneficial to identify the influence of various water bodies on different precipitation event. As Fig.2 shows, isotope compositions of most soil water are plotted in the bottom of the LMWL, and are close to the LMWL. The variable trends of D and $^{18}$O values of soil water are similar to that of precipitation. There is a strong linear relationship ($R^2 = 0.96$) for $\delta$D and $\delta^{18}$O between precipitation and soil water, which indicates that soil water is originally recharged by precipitation. The soil water is increasingly enriched in D and $^{18}$O, which are influenced by evaporation (Table 1). It has been reported that soil

water is normally more enriched because precipitation enters the soil, and mixes with antecedent soil water that has been modified by evaporation.

Vertical profiles of soil water $\delta$D-$^{18}$O in the Anjiagou River basin are shown in Fig.3. The surface layer (0-20 cm) showed a larger variations and higher $\delta$D-$^{18}$O values than the deeper layer. $\delta$D and $^{18}$O values of soil water are greatest at the surface layer (Fig.3,Table 1). $\delta$D -$^{18}$O values of shallow soil water at 0-10cmare-10.29‰ to -7.36‰,-72.71‰ to -35.45‰,

respectively. The $\delta$D-$^{18}$O values of shallower soil water at 10-20cm are also high (ranges from-8.08‰ to -3.57‰, and ranges from-67.27‰ to -49.70‰, respectively). The net effect of evaporation is the enrichment of heavy isotopes near the soil



surface. Soil profiles (Fig.3) show the increase in the D-$^{18}$Owith soil depths from the 0 to 20cm, suggesting the occurrence of evaporation. The large variation in isotope compositions of shallow depths (0-20cm depth) may be caused by precipitation and evapotranspiration effect. The correlation of the isotopic composition between soil water and precipitation is weak. These have been reported by many other studies (Gazis and Feng, 2004; Robertson and Gazis, 2006).

The $\delta^{18}$O and $\delta$D compositions of soil water at 40cm depths range from -11.35‰ to -7.09‰, and -74.21‰ to -54.06, respectively. Soil water from20 to 40cm depths is easily influenced by precipitation infiltration and evapotranspiration, which leads to the extremely variable $\delta$D and $\delta^{18}$O values of soil water (Fig.3). The impact of precipitation on soil water becomes weaker with increasing depths. The isotopic composition of soil water is strongly enriching in $\delta$D and $\delta^{18}$O due to less precipitation and evaporation in the study area. The $\delta$D and $\delta^{18}$O decrease exponentially with the increasing depths, and the variation trends are larger. Previous studies have also shown that evaporation decreases with the increasing soil depth, and its influence is generally within tens of centimeters. $\delta$D -$^{18}$O declined with depth to 60 cm but remained constant at deeper depths. When it reaches the 60cm depth, the variation trend tends to be stationary. The $\delta^{18}$O and $\delta$D compositions of soil water at 60cm depths ranged from -12.79‰ to -9.75‰, and ranged from -96.91‰ to -57.27, respectively. There is an abrupt peak value in the plant root zone. The isotopic fluctuations quickly disappear due to the dispersive effects of the deep root system.

However, the values of $\delta$D and $\delta^{18}$O are gradually decreasing from 60 to 80cm because the recharge of precipitation is decreasing, and the influence of antecedent isotope value of soil water is increasing. As the Fig.3 shows, D and $^{18}$O values of soil water at depths of 80 cm range from -11.53‰ to -8.92‰, and range from -78.82‰ to -62.80‰, respectively. Compared with the large variation in the $\delta$D and $\delta^{18}$O of surface soil water, soil water deeper than 80 cm is almost constant. The low value zone is at 100cm in the isotopic profiles of soil water, which is the deepest layer that recharged by precipitation. Variations of $\delta^{18}$O and $\delta$D of soil water in the deep layer are non-significant. Evaporation is larger than precipitation in the study area. Precipitation is difficult to infiltrate to 80 cm depth in the shrubland ecosystems, except in the high flow years. Therefore, it is difficult to cause the isotope fractionation of soil water blow 80cm.

**3.3 Variability in soil moisture content**



Values of soil moisture obtained by TDR in the Anjiagou River basin are shown in Fig.5. The variation of soil moisture content in the shallow soil layer (0-40 cm) is extremely large, ranging from 5.94 to 17.51%, which belongs to the variable greatly layer. The variation of soil moisture content at the 40-60 cm soil depth is relatively small, which belongs to the less active layer. The variation of soil moisture content is relatively stable at 60-100 cm soil depth, ranging from 17.78% to 29.47%, which belongs to the relatively stable layer. Soil water content also exhibits depth variation. Variability of water content is larger in the surface horizons more than 40 cm depth in the soil profile. The shallow soil layers are impacted by evaporation and precipitation recharge more than the deeper soil layer. Therefore, high evapotranspiration rate and precipitation recharge are the main factor controlling soil moisture especially the surface horizon.

As the soil moisture profile shows (Fig.4), soil moisture content is low in the shallow layer (0-20 cm), soil water content is relatively high at the active layer (40-60cm), and then tends to be stable. Figure 4 shows that soil moisture content is different in different sample sites, but soil moisture content is low under 60 cm soil depths in all the sample sites. Soil moisture content generally increases with the increase of soil depth in the variable greatly layer. Soil moisture content decreases with the increase of soil depth in the active layer, and then tends to be stable.

As Fig.4 shows, water content increases and soil water $\delta^{18}$O-D values increase in the shallow layer (0-40 cm), which are impacted by evapotranspiration. This is the dominant process for most of the sites. Mean value of the soil moisture content increases with depth in the soil profile (40-60cm depth), while soil water $\delta^{18}$O-D values decrease with depth, which mixes with antecedent moisture. Low water content and more positive $^{18}$O-D values (Fig.4) occur at the deep soil layer (below 60cm depth), which is impacted by recharge. Single precipitation is difficult to infiltrate to deeper than 80 cm, when drought lasts for a long time, especially when soil water content lower than 20%. Extremely high values of soil water content at the depth of 60 cm are observed in Anjiagou River basin. The higher water content and more negative d$^{18}$O values are observed at the active layer (40-60 cm) (Fig.4).The variability of water content and the $^{18}$O-D values are relatively larger for the surface horizons than deeper in the soil profile. The profile of soil moisture content is highly responsive to evaporation and precipitation. The average water content and soil water $\delta^{18}$O-D values are representative of the trends found in most of the sample sites.

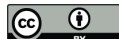



## 4. Discussion

### 4.1 The migration process of soil water in the unsaturated zone

Precipitation infiltration and evapotranspiration in the vertical direction are the main processes of the water cycle, which plays an important role in the transformation process of water cycle in the unsaturated zone. Different migration mechanisms of soil water result in different isotopic profiles. Therefore, isotope $\delta D$ and $\delta^{18}O$ of soil water and precipitation can be obtained to identify soil water migration processes such as infiltration, evaporation, transpiration and percolation through the soil waters at different depths. These different isotopic profiles may occur as a result of several processes, including evaporation, change in isotopic composition of precipitation and mixing of new and old water. While evaporation is widely recognized, we believe that mixing is also important factors for controlling the isotopic composition of soil water at these sites. These are possibly caused by the infiltration of summer rainwater through the soil layer. Generally, there are two major infiltration mechanisms (piston-type flow and preferential flow) that can be identified by comparing the isotope compositions of precipitation and soil water.

### 4.1.1 Evidences for piston-type flow

The $\delta D$ and $\delta^{18}O$ content of soil water at different depths are shown in Fig.4. By comparing the isotopic compositions of precipitation and soil water, the mechanisms of soil water movement can be identified. Most of the sampling sites in Anjiagou River basin (except 14-7-20 and 14-9-24 sites), the isotopic composition of soil water in shallow layer are enriched in D and $^{18}O$ due to evaporation. The isotopic composition of soil water changes with depth in the soil profile, especially after the continuing precipitation between August 18 and August 20, 2014. Precipitation is infiltrating and completely mixing with free water. The vertical trend in $\delta D$ and $\delta^{18}O$ profile of soil water is simple. There is an abrupt peak value in the isotopic profiles, which suggests that the older water is pushed downward by the new water and infiltrates to the deep soil. That is, while some soil water remains stationary, the mobile soil water successively displaces pre-existing mobile soil water pushing it downward. Piston-type flow after an isotopic distinct rainfall event results in an abrupt ''isotopic front'' within the soil (Gehrels et al., 1998; Song et al., 2009).This evidence for mixing, abrupt changes in the $\delta D$ and $\delta^{18}O$ of soil water with depth




in the soil profile indicates that piston-flow has occurred. A relatively flat pattern is observed, slightly enriched in accordance with precipitation. δD and $\delta^{18}$O declined with depth to 60 cm but remained constant at deeper depths. The older

soil water is pushed by newer rainwater downward. Precipitation infiltration exhibits piston infiltration characteristics (Zimmerman, 1967).

### 4.1.2 Evidences for preferential flow

At the 14-7-20 and 14-9-24 samples sites, the vertical trend of δD and $\delta^{18}$O in the soil profile indicates a uniform infiltration process (Fig.5). There are two abrupt peak values in the isotopic vertical profiles of soil water. The soil water at depths

between 20 cm and 40 cm decrease quickly, which is caused by the strong transpiration effect of plants roots. Abrupt change occurred at 20 cm-40 cm depth of the isotopic profile where the soil water moisture increased quickly (range from -14.97‰ to -12.78‰, range from -107.93‰ to -101.21‰, respectively).There are two continuous rainfall events on20July 2014 and 24 September 2014, which infiltrating uniformly into deep depths, as shown by the reducing δD and $\delta^{18}$O values with time after the precipitation (Fig.4). The data do not show the obvious correlation with depth at the14-7-20 and 14-9-24 samples

sites in Anjiagou River basin (Fig.4). This is mainly caused by the mixture effect occurred during the process of infiltration. That is to say, the newer water, which did not replace the old water totally during the process of infiltration, mixed with the old water stored in the soil.

     The second peak of $\delta^{18}$O and δD values of soil water is at the depth of 80 cm (range from -12.41‰ to -10.75‰, range from -98.87‰ to -82.77‰, respectively), which is caused most likely by macropore flow. The non-existence of a significant

difference between the isotopic content of precipitation and soil water suggests that the infiltrating summer precipitation passes the root zone quickly and contributes to the soil water recharge. The deep soil water is recharged by precipitation in the form of preferential flow, which passes the soil porosity and quickly reaches the deep layer, and do not mix with older water. Variations of $\delta^{18}$O and δD of deep soil water are only by mixing, and rarely by evaporation. A number of studies documenting tracer movement in the soil column have pointed out the existence of preferential fluid flow, which can be

caused by cracks (in dried soils), macropores (Beven and Germann, 1982), plant roots, earthworm burrows, etc. (Vincent et al., 2001; Bronswijk et al., 1991). However, the preferred flow is rarely found in the Loess Plateau, except that there are

macrospores, which is caused by plant root system or animal invasion, etc. The deep soil water is quickly recharged by

precipitation in the form of preferential flow through observing the excavated soil profile. The macrospores can be found in

the 100-200cm soil layer in the study area, which are caused by plant root or animal activity. The cracks in the Loess Plateau

can also provide the important path for preferential flow. This is supported by the variations of soil water content. The soil

water content at 80 cm depth increases quickly. The soil water content ranges from 34.5 to 36.8%. At corresponding depths

of the isotope profile, the isotope values are relatively depleted. Significant soil water content increase in the deep soil is

similar to variability of isotopic composition of soil water. The isotope profile can provide some evidence for preferential

flow.

**4.2 water origin and contribution of each potential water sources are uptake by plants**

**4.2.1 Water origin of water sources are uptake by plants**

Zimmermann et al. (1967) studied the effect of transpiration with an experiment in which the root water uptake of various

plant species is monitored, and no fractionation is found. Allison et al. (1985) tested a much greater fraction of root water

uptake, no conclusive proof of fractionation by root water uptake is found. The isotope fractionations are not caused when

soil water is uptaken by roots of plant. By comparing the $\delta D$ and $\delta^{18}O$ of plant water and each water sources, the water origin

and contribution of each component for xylem could be identified.

The isotopic composition of soil layer is similar to the xylem water is confirmed by comparing $\delta D$ and $\delta^{18}O$ of xylem

water and soil water, which reflect the signatures of soil at the depth of soil water uptake by plants. All the soil and xylem

samples are taken simultaneously in Anjiagou River basin. The isotopes of $\delta D$ and $\delta^{18}O$ are conservative and do not react

with clay minerals and other soil materials. During the process of transpiration, the soil water content changes, but the

isotopic composition of soil water remains constant. The isotopic composition of plant xylem water is a mixture of soil water

at different soil depths. Through analyzing the $\delta D$ and $\delta^{18}O$ in xylem water of *caraganakorshinshiikom,* and the $\delta D$ and $\delta^{18}O$

in potential source such as soil water and precipitation in Anjiagou Gully basin (Table 2). Comparing the signatures $\delta D$

and$\delta^{18}O$of both soil water and xylem water with a simple graphical interference approach indicates a distinct difference

between both isotopes. Graphical "best match" approach is suitable to illustrate water uptake depth as plants often withdraw

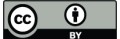

water from more than one distinct soil depth. The $\delta$D and $\delta^{18}$O in xylem water ranged from -11.64‰ to -7.95‰, and from -81.62‰ to -55.21‰, respectively (Fig.6, Table 2). Plant Xylem water $\delta$D and $\delta^{18}$O for both species is similar to that of soil water at 40-60cm depth (Fig.6).The results show that isotope composition of xylem water of various plants is different. It indicates that soil water at the 40-60cm depth is mainly used by *caraganakorshinshiikom*.

**4.2.2 The contribution of each potential water sources to plants water**

Hooper et al. (1990) and Christophersen and Hooper (1992) introduced the end-member mixing analysis (EMMA), which is an often-used method for analyzing possible source area contributions to flow. Multiple source mixing models (Parnell et al. 2010; Phillips et al.2003) account for water uptake from more than one discrete soil layer and weigh the importance of certain layers for water uptake by incorporating e.g. soil water potentials into the calculation. In this study, the mixing model is used to identify potential source areas and mixing processes, and to quantify the contribution of each end member using mixture fractions. The conservative elements $\delta$D and $\delta^{18}$O are utilized as tracers to calculate mixing amounts.

It is more likely that plant xylem water is a mixture of soil water from several soil depths. Multiple source mass-balance mixing models of proportional contributions of plant xylem water (%) (Parnell et al. 2010) is used to evaluate the contributions of each potential water source (Fig.6).The results show that,soil water from the surface horizons (20-40cm) contributed to 8%-21% of plant xylem water, while soil water at 40-60cm soil depth is the largest component of plant xylem water (range from 68% to 83%), soil water below 60 cm depth contributed 10%-26% to plant xylem water, and only 0%-18% is contributed directly from precipitation. Water source is dominated by soil water at a depth of 40-60 cm.

**5. Conclusions**

The following conclusions can be drawn from the present study:

(1) $\delta^{18}$O and $\delta$D enrichment in the shallow (< 20 cm depth) soil water, observed in most soil profiles, is due to evaporation. The isotopic composition of xylem water is close to that of soil water at a depth of 40-60 cm. These values reflect soil water signatures associated with shrubland uptake at a depth of 40-60 cm.

(2) A sharp isotopic front at approximately 40cm depth observed shortly after an isotopic distinct rainfall event suggests

that infiltration into soil occurred as piston-type flow with newer water pushing older water downward in the soil profile.

Soil waters are recharged from precipitation. The soil water migration is dominated by piston-type flow in the study area, but

rarely preferential flow, except that there are macrospores in the Loess Plateau, which is caused by plant root or animal

invasion, etc. Water migration exhibited a transformation pathway from precipitation to soil water to plant water.

(3)Soil water from the surface horizons (20-40cm) contributed to 8%-21% of plant xylem water, while soil water at

40-60cm soil depth is the largest component of plant xylem water (range from 68% to 83%), soil water below 60 cm depth

contributed 10%-26% to plant xylem water, and only 0%-18% is contributed directly from precipitation. Water source is

dominated by soil water at depth of 40-60cm.

### Acknowledgements

This research is supported by National Natural Science Foundation of China (41390464), China Postdoctoral Science

Foundation Funded Project (2014M550095), Science and Technology Major project of Shanxi Province (20121101011).The

authors are grateful to experimental research station and all participants in the field for their contributions to the progress of

this study. We also express our appreciation to the anonymous reviewers of the manuscript.

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



**Figure captions in sequence**

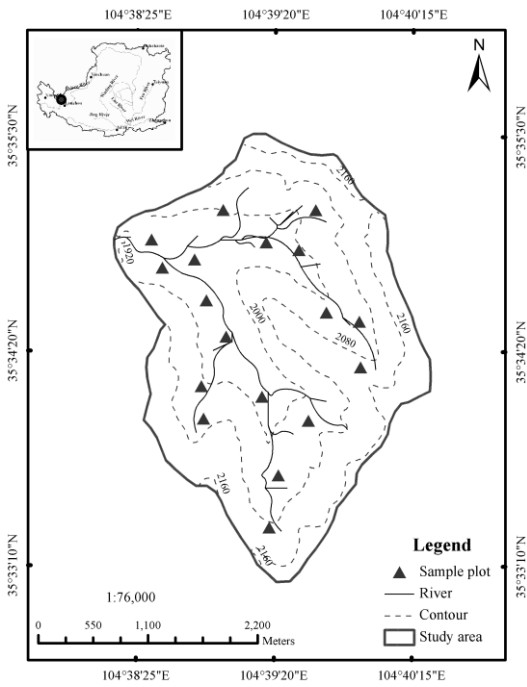

Fig.1 Location of the sampling sites in the study area





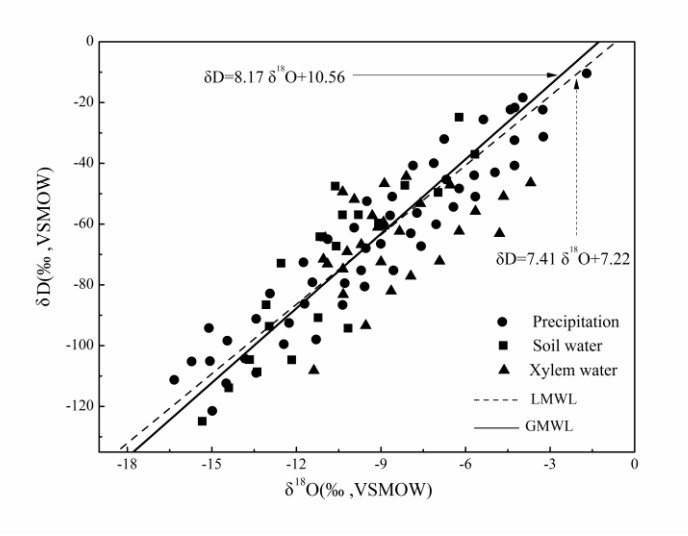


Fig.2 Plot of δD versus $\delta^{18}$O for the various samples in the study area

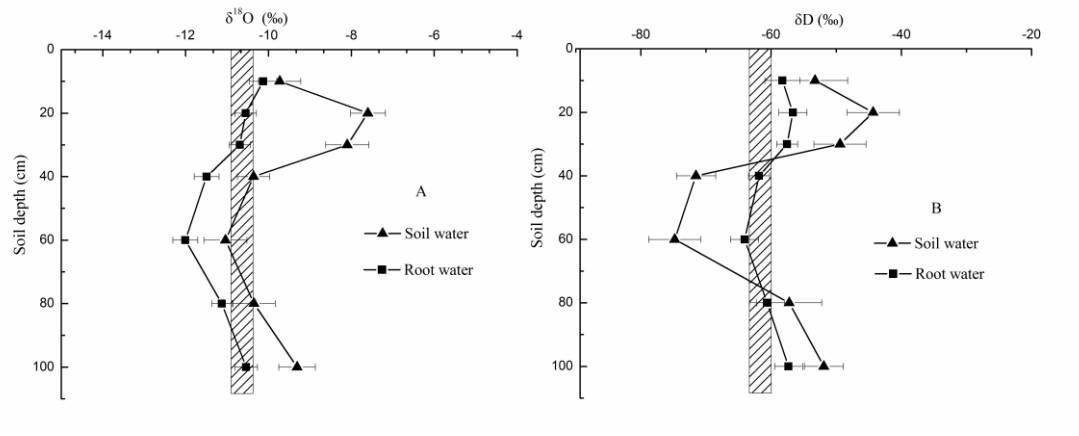

Fig.3 Isotope profiles of $\delta^{18}$O and δD (piston-type flow) in soil water and xylem water. Values are meas± standard deviation

(*n*=27)






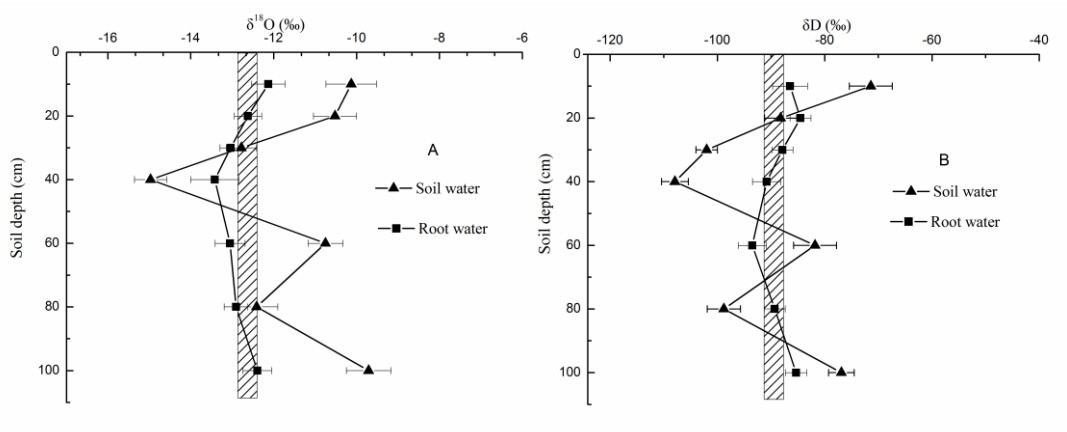

Fig.4 Isotope profiles of δ¹⁸O and δD (preferential flow) in soil water and xylem water. Values are meas± standard deviation

(*n*=27)






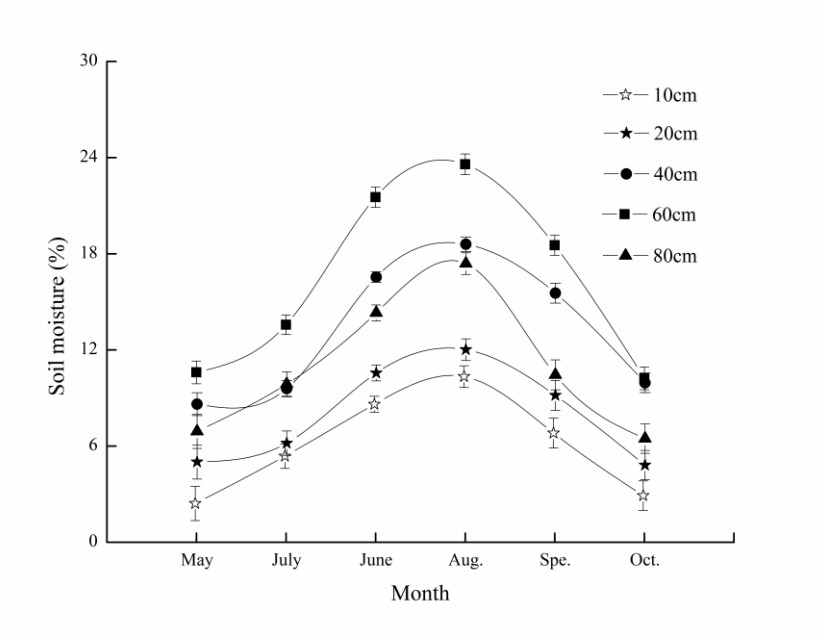

Fig.5The variation of soil moisture content at the differentsoil depth

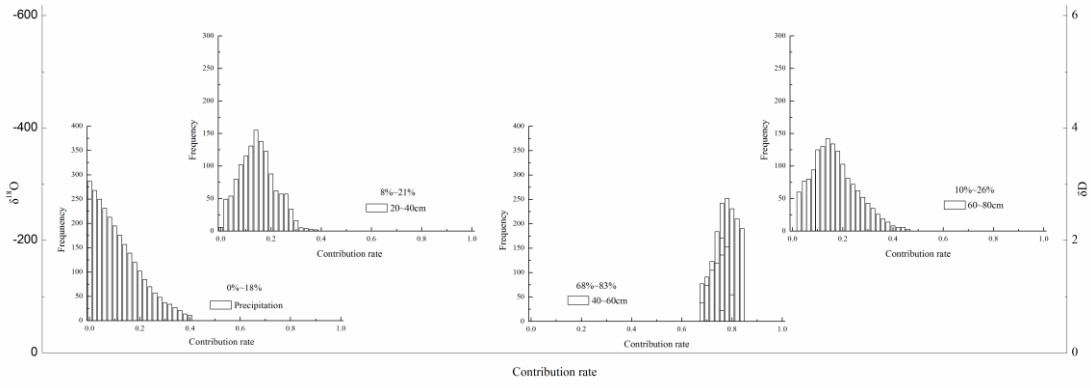

Fig.6The contributions of each potential water sources to plants





**Table captions in sequence**

Tab.1 Statistic characteristics of isotopic compositions for precipitation and soil water and in the study area

| Type | Depth (cm) | $\delta^{18}O.‰$ | | | | | $\delta D.‰$ | | | | |
|------|------------|---|-----|-----|------|-----------|---|-----|-----|------|---------|
|      |            | N | Min | Max | Mean | Std.Devj. | N | Min | Max | Mean | Std.Dev |
| Precipitation | | 18 | -16.35 | -2.19 | -8.26 | 1.9 | 18 | -118.36 | -22.32 | -55.15 | 5.7 |
| Soil Water | 10 | 27 | -10.29 | -7.36 | -9.73 | 0.7 | 27 | -67.27 | -49.70 | -53.24 | 3.9 |
| | 20 | 27 | -8.80 | -3.57 | -7.20 | 1.2 | 27 | -72.71 | -44.32 | -48.32 | 4.1 |
| | 40 | 27 | -11.35 | -7.09 | -10.36 | 0.6 | 27 | -74.21 | -61.53 | -71.31 | 1.7 |
| | 60 | 27 | -12.79 | -9.75 | -11.24 | 0.9 | 27 | -86.91 | -74.8 | -76.80 | 2.4 |
| | 80 | 27 | -11.53 | -8.92 | -10.35 | 0.5 | 27 | -78.82 | -57.22 | -62.42 | 1.8 |
| | 100 | 27 | -10.75 | -8.34 | -9.31 | 0.6 | 27 | -78.11 | -51.89 | -64.89 | 1.5 |






Tab.2 Statistic characteristics of isotopic compositions for root water and xylem water in the study area

| Type | Depth (cm) | δ¹⁸O.‰ | | | | | δD.‰ | | | | |
|---|---|---|---|---|---|---|---|---|---|---|---|
| | | N | Min | Max | Mean | Std.Devj . | N | Min | Max | Mean | Std.Dev |
| xylem water | | 27 | -11.64 | -7.95 | -10.61 | 0.66 | 27 | -81.62 | -55.21 | -69.94 | 4.9 |
| | 10 | 27 | -14.21 | -5.12 | -8.34 | 0.64 | 27 | -62.39 | -52.99 | -62.37 | 5.4 |
| | 20 | 27 | -10.27 | -5.97 | -9.21 | 1.5 | 27 | -56.21 | -44.18 | -59.38 | 3.3 |
| Root water | 40 | 27 | -11.79 | -6.91 | -10.1 | 0.89 | 27 | -73.20 | -61.56 | -67.05 | 4.2 |
| | 60 | 27 | -12.47 | -8.08 | -10.6 | 0.77 | 27 | -83.49 | -65.95 | -73.14 | 2.9 |
| | 80 | 27 | -11.99 | -8.92 | -9.7 | 0.81 | 27 | -81.18 | -57.72 | -66.75 | 1.7 |
| | 100 | 27 | -9.80 | -7.3 | -8.9 | 0.64 | 27 | -70.57 | -59.27 | -60.62 | 2.5 |