# Peer review of "Soil water migration in the unsaturated zone of semi-arid region in China from isotope evidence"

_Hydrology and Earth System Sciences, 2016_

## Referee Comment (RC1) · Anonymous Referee #1 · 5 Dec 2016

The subject of this manuscript is interesting and worthy of publication. I believe that the topic will be of broad interest to readers. In my opinion, the key contribution of the manuscript is that it aims to investigate soil water migration processes using isotopes, integrated with sampling in the field, experimental observation and laboratory analysis, and to integrate data from several different compartments of the hydrological system, i.e. precipitation, soil water, plant xylem and root etc.

However, I think some minor modifications are required before the manuscript's publication.

2.2 Sample collection and field experiment "Samples of precipitation, soil, xylem, stem, leaf and root were collected in the study area." "Stem samples (about 0.5 to 2 cm

diameter and 5 cm length) were obtained." While no linkage of analysis about stem and leaf was found with the previous and the subsequent discussions. This paragraph should be moved and edited.

4.1.1 Evidences for piston-type flow There are two continuous rainfall events on 20 July 2014 and 24 September 2014, which infiltrating uniformly into deep depths, as shown by the reducing $\delta$ D and $\delta$ 18 O values with time after the precipitation.

Here how many days did the two continuous rainfall events last respectively? What about the precipitation values? Please give specific explanation.

4.2.1 Water origin of water sources are uptake by plants "The results show that isotope composition of xylem water of various plants is different. In the paper, what part can explain this point? There is no such comparative analysis about different plants, so please give further explanation or delete.

In addition, please give explanation about the reason of choosing 130cm as the maximum sampling depth. What about groundwater? Why no discussion in this study?

---

## Referee Comment (RC2) · L. Bing (Referee) · 14 Dec 2016

The purpose of this paper is to probe into the migration process soil water, and identify each potential water source, which is well justified. There are also good data sets. To my knowledge, this would be the comprehensive study of soil water migration in the unsaturated zone in the western China. It does look like the data are very good and useful, especially the isotopic data. Publication of this study could advance our understanding of alpine hydrology and also be of interest to hydrologists. There are some problems with the paper, and it will be revised to get it for publication in the journal. They are as follows:

(1) Add an inset map (e.g., a small China map) showing where the basin is relative to

Beijing and Dingxi.

(2) On page 13 it is stated that isotope composition of xylem water of various plants is different. Soil water at the 40-60cm depth is mainly used by caraganakorshinshiikom. Groundwater? Precipitation? Why? I am not saying that the recharge is not possible but the stronger hydrogeological background is needed for it to be plausible. The Area description, especially the Hydrological Background, is also required.

(3) Vacuum distillation was considered as a reliable and accepted method for soil water and pant water abstraction in this paper. Please give a description on detailed procedure, such as time and temperature selection.

(4) On page 12, "Through analyzing the $\delta$D and $\delta$ 18 O in xylem water of caraganakorshinshiikom, and the $\delta$D and $\delta$ 18 O in potential source such as soil water and precipitation in Anjiagou Gully basin".Here what is the meaning of "in Anjiagou Gully basin"ïij§

(5) How about terrain and vegetation on soil water?

————————————————

---

## Referee Comment (RC3) · Anonymous Referee #3 · 18 Dec 2016

This paper examined how water infiltration, evaporation, and transpiration affected the isotope distribution in soil profile. They used water isotopes assisted by the water content measurements. They also used water sourcing approach to identify how much water come from different soil depths and precipitation. Stable isotope is very effective for this research. A lot of data is acquired, and the new knowledge is obtained. I suggest some modifications before publication.

The following are my suggestions.

1, To add description of soil and vegetation in the study area;

2, To add a description of soil water sampling time;

[Figure]

3, Keywords should be hydrogen and oxygen stable isotope;

4, How many precipitation samples were there to define LMWL? Please add sample number.

5, To increase the new relevant references.

---

## Referee Comment (RC4) · Anonymous Referee #4 · 19 Dec 2016

Soil moisture for vegetation growth is very important in the Loess Plateau. Soil water migration is essential to describe the movement of salt, carbon, nitrogen and other nutrient. Although there are many studies on the Soil moisture, there are less works on the mechanism of soil water migration in the unsaturated zone in the Loess Plateau of arid region in Chinaï¡ÿbased on successive observations and isotopes. The scientific issue of this paper is sure, research idea is clear, research method is correct, and research results are reliable. From my point of view, the work is well-done and provides reliable results to soil water migration in arid region of China.

1. Adding relevant detailed introduction about soil and hydrology in the research area.

2. As mentioned in the paper, "Soil moisture content was determined by oven dry-

ing method simultaneously" How much about the depth of soil sample? Does every depth be concluded respectively? Does the measured depth be in accordance with that of time domain reflectometry ? 3. Please explain the reason for choosing caraganakorshinshiikom as the object of study, and illustrate its characteristics in the part of research area summary. 4.there are many mentioned in the paper, such as"Plant Xylem" "The isotopic composition of soil layer is similar to the xylem water is confirmed by comparing $\delta$D and $\delta$18O of xylem water and soil water, which reflect the signatures of soil at the depth of soil water uptake by plants."" soil water from the surface horizons (20-40cm) contributed to 8%-21% of plant xylem water," plant roots. Is it the xylogen and root system of a certain kind of plant? Caraganakorshinshiikom? In general, this is a well-written paper containing reliable results which merits publication.

---

## Referee Comment (RC5) · Anonymous Referee #5 · 21 Dec 2016

The authors have tried to do some fundamental research based on field survey and water samples analyses. The study sampled soil profile water stable isotopes of soil, and discussed soil water-plant water recharge sources. This study provides some useful information for soil moisture dynamics and stable characteristics. So it is significant. However, I could recommend the paper for publication after the following issues have been addressed by the authors:

P1 L19, D is used in the text for the first time, so please state this is Deuterium.

P4 , L 80ãÃĄ"groundwater in the unsaturated zone" There is no relevant analysis about groundwater in the whole passage, so it should be deleted. P7 , L52-53ãÃĄ P8 ,167 "The measured D and18 O of soil water range from -72.42‰ to -37.05‰ and -11.74‰

to -3.57‰ respectively." Add the little "delta" before 18O such as $\delta$ 18O.

p10, L 230-233ïijŇ" Most of the sampling sites in Anjiagou River basin (except 14-7-20 and 14-9-24 sites), the isotopic composition of soil water in shallow layer are enriched in D and 18O due to evaporation. The isotopic composition of soil water with depth in the soil profile. " Here the changes should be specifically explained.

p12, L 261-264ïijŇ"the preferred flow is rarely found in the Loess Plateau, except that there are which is caused by plant root system or animal invasion, etc." "The macrospores can be found in the 100-200cm soil layer in the study area, which are caused by plant root or animal activity." Here "plant root or animal activity" should be changed into "plant root or animal invasion", and exact illustration is needed.

P13 , L 292ãø"often-used" Here "often-used" should be changed into "often used".

---

## Short Comment (SC1) · 4 Jan 2017

The literature search is not current enough and misses some important recent papers:

L44: This is completely wrong. Read and include the references of Gaj et al., 2017; Oerter et al., 2014, Meibner et al., 2014; Lin and Horita, 2016.

L45-46: Need to include references to support this claim. I suggest the authors read and include citations from Sprenger et al. 2016.

L115: This needs some justification. There must be some discussion (somewhere in the paper, but probably not in the methods section) of the many papers that cite problems with the vacuum extraction process and variability in its results, see Araguas-

Araguas et al., 1995; Orlowski et al., 2016a; Orlowski et al, 2016b. Details on the vacuum distillation methods need to be included.

References that need to be included:

Araguas-Araguas, L., Rozanski, K., Gonfiantini, R., Louvat, D., 1995. Isotope effects accompanying vacuum extraction of soil water for stable isotope analyses. J Hydrol, 168: 159-171.

Gaj, M. et al., 2017. Mineral mediated isotope fractionation of soil water. Rapid Communications in Mass Spectrometry, 31(3): 269-280.

Lin, Y., Horita, J., 2016. An experimental study on isotope fractionation in a meso-porous silica-water system with implications for vadose-zone hydrology. Geochimica et Cosmochimica Acta, 184: 257-271.

Meißner, M., Köhler, M., Schwendenmann, L., Hölscher, D., Dyckmans, J., 2014. Soil water uptake by trees using water stable isotopes ($\delta$2H and $\delta$18O)$-$ a method test regarding soil moisture, texture and carbonate. Plant and Soil, 376(1-2): 327-335.

Oerter, E. et al., 2014. Oxygen isotope fractionation effects in soil water via interaction with cations (Mg, Ca, K, Na) adsorbed to phyllosilicate clay minerals. J Hydrol, 515: 1-9.

Orlowski, N., Breuer, L., McDonnell, J.J., 2016a. Critical issues with cryogenic extraction of soil water for stable isotope analysis. Ecohydrology, 9: 3-10.

Orlowski, N., Pratt, D.L., McDonnell, J.J., 2016b. Intercomparison of soil pore water extraction methods for stable isotope analysis. Hydrol Process, 30(19): 3434-3449.

Sprenger, M., Leistert, H., Gimbel, K., Weiler, M., 2016. Illuminating hydrological processes at the soil - vegetation - atmosphere interface with water stable isotopes. Reviews of Geophysics.

---

## Author Comment (AC1) · 1 Feb 2017

**hess-2016-555-RC1**

The subject of this manuscript is interesting and worthy of publication. I believe that the topic will be of broad interest to readers. In my opinion, the key contribution of the manuscript is that it aims to investigate soil water migration processes using isotopes, integrated with sampling in the field, experimental observation and laboratory analysis, and to integrate data from several different compartments of the hydrological system, i.e. precipitation, soil water, plant xylem and root etc.

However, I think some minor modifications are required before the manuscript's publication.

2.2 Sample collection and field experiment "Samples of precipitation, soil, xylem, stem, leaf and root were collected in the study area." "Stem samples (about 0.5 to 2 cm diameter and 5 cm length) were obtained." While no linkage of analysis about stem and leaf was found with the previous and the subsequent discussions. This paragraph should be moved and edited.

**Response: Done.** We have revised it. The statement of"stem and leaf" has been removed. The analysis about stem and leaf will be the research point in the future.

4.1.1 Evidences for piston-type flow there are two continuous rainfall events on 20 July 2014 and 24 September 2014, which infiltrating uniformly into deep depths, as shown by the reducing D and $^{18}$O values with time after the precipitation. Here how many days did the two continuous rainfall events last respectively? What about the precipitation values? Please give specific explanation.

**Response: Done.** We have revised it. The rainfall event on 20 July 2014, with the precipitation of 25mm, lasted for three days. The rainfall event on 24 September 2014, with the precipitation of 16mm, lasted for two days.

4.2.1 Water origin of water sources are uptake by plants "The results show that isotope composition of xylem water of various plants is different". In the paper, what part can explain this point? There is no such comparative analysis about different plants, so please give further explanation or delete. In addition, please give explanation about the reason of choosing 130cm as the maximum sampling depth.

What about groundwater? Why no discussion in this study?

**Response: Done.** We have revised it. In this paper, our discussion focuses on water origin of caraganakorshinshiikom xylem. Therefore, we delete the statement of "The results show that isotope composition of xylem water of various plants is different."

Characterized by dry climate, less precipitation, more evaporation and thicker soil layers, and groundwater buried deeper, groundwater in the study area is difficult to use due to the depth of water table in the northwestern Loess Plateau. The thickness of the soil layer in vadose zone is typically Tens of meters, even a few hundred meters. Many researchers have shown that groundwater in the study area has no effect on plants. Therefore in this study the influence of groundwater does not exist. Thus, soil water is almost the only water resource in the study area. *Caraganakorshinshiikom* in the study area are artificial planting, old small, root system distribution within the 100 cm. There is little more than 100 cm depth. Therefore, we choose 130cm as the maximum sampling depth.

---

## Author Comment (AC2) · 1 Feb 2017

The purpose of this paper is to probe into the migration process soil water, and identify each potential water source, which is well justified. There are also good data sets. To my knowledge, this would be the comprehensive study of soil water migration in the unsaturated zone in the western China. It does look like that the data are very good and useful, especially the isotopic data. Publication of this study could advance our understanding of alpine hydrology and also be of interest to hydrologists. There are some problems with the paper, and it will be revised to get it for publication in the journal. They are as follows:

(1) Add an inset map (e.g., a small China map) showing where the basin is relative to Beijing and Dingxi.

**Response: Done.** We have revised it. We have added an inset map (China map), which shows where the basin is relative to Beijing and Dingxi.

(2) On page 13 it is stated that isotope composition of xylem water of various plants are different. Soil water at the 40-60cm depth is mainly used by caraganakorshinshiikom. Groundwater? Precipitation? Why? I am not saying that the recharge is not possible but the stronger hydrogeological background is needed for it to be plausible. The Area description, especially the Hydrological Background, is also required.

**Response: Done.** We have revised it.

Characterized by dry climate, less precipitation, more evaporation and thicker soil layers, and groundwater buried deeper, groundwater in the study area is difficult to use due to the depth of water table in the northwestern Loess Plateau. The thickness of the soil layer in vadose zone is typically Tens of meters, even a few hundred meters.

The climate is semi-arid, with an average annual temperature of 6.3℃,annual accumulated temperature of 2239℃, extreme maximum temperature 34.3℃, extreme minimum temperature of -27.1 ℃ in the study area. The annual mean precipitation is 420mm. The annual mean evaporation is

1510mm. The aridity index is 1.15. Precipitation is low and unevenly distributed in temporal and spatial. The summer rainfall accounted for over 60% of the annual precipitation. This area is a part of the typical semi-arid in the hilly and gully region of Loess Plateau, with the altitude ranging from 1900m to 2250m.

The watershed area is 8.91km$^2$, which belongs to the hilly and gully region of Loess Plateau. Gully density is 3.14km/km$^2$, and the ditch depth ranges from 30 to 50m. Soil type is yellow loessal soil and saline soil, and the average thickness ranges from 40 to 60m. The soil density of soil layer ranges from 1.1 to 1.4 g/cm$^3$, average soil porosity is 55%. Soil structure has a vertical joint, and the nature of soil is loose and its wet collapsibility is serious. The grassland and shrubland ecosystems are the most extensively dominant ecosystems in the Anjiugou river basin. As it is a representative area of Loess Plateau area, the Anjiagou River basin is a suitable area for soil water study in semi-arid region.

The study area has broken terrain and serious soil erosion, with the terrain being loess long beam and terraces, and gully valley landscape. Geological structure is the uplift zone between the eastern part of Qilian fold system and the west Qinling fold system, at an altitude of 1700 m ~ 2580 m, with the gully density being 3 ~ 5 km/km$^2$, ditch slope being 5 ~ 10%, and the mountain slope being generally 20 to 50. The sunny slope is steep, while the shade slope is relatively flat.

The soil parent material is quaternary eolian loess, and the zonal soil mainly is yellow spongy soils, sierozem, which belongs to the typical semiarid loess hilly-gully region. It has soft soil, homogeneous structure, thicker soil layer, good water performance, and the widest distribution. The average thickness is 40 ~ 60 m. Clay soil is between 33.12% ~ 42.17%, organic matter content is between 0.37% ~ 1.34%, soil bulk density is 1.17 g/cm$^3$, wilting moisture content is 7.3%, and the saturated moisture content is 21.9% at the 0 ~ 20 cm. The soil bulk density is 1.09~1.36g cm$^{-3}$, and the porosity is 50% ~ 55% at the 2 m soil layer. The soil has vertical joint and strong collapsibility, so the soil erosion is easily happened, and the soil erosion modulus is 5000 t/km$^2 \cdot$ a.

(3) Vacuum distillation was considered as a reliable and accepted method for soil water and pant water abstraction in this paper. Please give a description on detailed procedure, such as time and temperature selection.

**Response: Done.** We have revised it.

Water was extracted from soil, leaf, branch, xylem and root by cryogenic vacuum distillation method. The moisture in the soil or plants under the condition of vacuum (vacuum below60MT), was heated by heat set to 105 ℃ after evaporation. Water vapor of evaporation in -50 ℃ (liquid nitrogen) was collected with frozen water collecting pipe (top-down frozen, in order to increase the collecting rate), and the precision extract of δD and $\delta^{18}$ are ±3‰ and ±0.3‰, respectively.

(4) On page 12, "Through analyzing the _D and _ 18 O in xylem water of caraganakorshinshiikom, and the _D and _ 18 O in potential source such as soil water and precipitation in Anjiagou Gully basin". Here what is the meaning of "in Anjiagou Gully basin"ïïj§

**Response: Done.** We have revised it. The correct expression should be "in Anjiagou River basin"

(5) How about terrain and vegetation on soil water?

**Response: Done.** We have revised it.

The vegetation type belongs to arid grassland vegetation type, with less distribution of natural arbor. The grassland and shrubland ecosystems were the most extensive dominant ecosystems. Woodland area is less; with most being open forest land. The area of the crown density being greater than 0.2 are only Caragana intermedia and *Pinus tabulaeformis*, and species are single. The distribution of natural herbs mainly are Ben's s. grandis, thyme, cold pole, camel peng, with vegetation coverage 10 ~ 20%. Natural coverage is commonly 25% ~ 35% in the sunny slope, 50% ~ 60% in the shade slope. Natural vegetations are mainly compositae Asteraceae, Leguminosae, Gramineae, etc. The vegetation is sparse, and species are relatively poor because of long-term influence by human activities. The artificial forest vegetations mainly are Caragana intermedia, Hippophae rhamnoides, Pinus tabulaeformis, Platycladus orientalis, Stipa bungeana, etc.

The study area has broken terrain and serious soil erosion, with the terrain being loess long beam and terraces, and gully valley landscape. Geological structure is the uplift zone between the eastern part of Qilian fold system and the west Qinling fold system, at an altitude of 1700 m ~ 2580 m, with the gully density being 3 ~ 5 km/km$^2$, ditch slope being 5 ~ 10%, and the mountain slope being generally

20 to 50. The sunny slope is steep, while the shade slope is relatively flat.Clay soil is between 33.12% ~ 42.17%, organic matter content is between 0.37% ~ 1.34%, soil bulk density is 1.17 g/cm$^3$, wilting moisture content is 7.3%, and the saturated moisture content is 21.9% at the 0 ~ 20 cm. The soil bulk density is 1.09~1.36g cm$^{-3}$, and the porosity is 50% ~ 55% at the 2 m soil layer. The soil has vertical joint and strong collapsibility, so the soil erosion is easily happened, and the soil erosion modulus is 5000 t/km$^2 \cdot$ a.

---

## Author Comment (AC3) · 1 Feb 2017

The comment was uploaded in the form of a supplement:
http://www.hydrol-earth-syst-sci-discuss.net/hess-2016-555/hess-2016-555-AC3-supplement.pdf
* * *

---

## Author Comment (AC4) · 1 Feb 2017

**hess-2016-555-RC3**

This paper examined how water infiltration, evaporation, and transpiration affected the isotope distribution in soil profile. They used water isotopes assisted by the water content measurements. They also used water sourcing approach to identify how much water come from different soil depths and precipitation. Stable isotope is very effective for this research. A lot of data is acquired, and the new knowledge is obtained. I suggest some modifications before publication.

The following are my suggestions.

1, To add description of soil and vegetation in the study area;

**Response: Done.** We have revised it.

The soil parent material is quaternary eolian loess, and the zonal soil mainly is yellow spongy soils, sierozem, which belongs to the typical semiarid loess hilly-gully region. It has soft soil, homogeneous structure, thicker soil layer, good water performance, and the widest distribution. The average thickness is 40~60 m. Clay soil is between 33.12% ~ 42.17%, organic matter content is between 0.37% ~ 1.34%, soil bulk density is 1.17 g/cm$^3$, wilting moisture content is 7.3%, and the saturated moisture content is 21.9% at the 0 ~ 20 cm. The soil bulk density is 1.09~1.36g cm$^{-3}$, and the porosity is 50% ~ 55% at the 2 m soil layer. The soil has vertical joint and strong collapsibility, so the soil erosion is easily happened, and the soil erosion modulus is 5000 t/km$^2 \cdot$ a.

The vegetation type belongs to arid grassland vegetation type, with less distribution of natural arbor. The grassland and shrubland ecosystems were the most extensive dominant ecosystems. Woodland area is less; with most being open forest land. The area of the crown density being greater than 0.2 are only Caragana intermedia and *Pinus tabulaeformis*, and species are single. The distribution of natural herbs mainly are Ben's s. grandis, thyme, cold pole, camel peng, with vegetation coverage 10 ~ 20%. Natural coverage is commonly 25% ~ 35% in the sunny slope, 50% ~ 60% in the shade slope. Natural vegetations are mainly compositae Asteraceae, Leguminosae, Gramineae, etc. The vegetation is sparse, and species are relatively poor because of long-term influence by human activities. The artificial forest vegetations mainly are Caragana intermedia, Hippophae rhamnoides, Pinus tabulaeformis, Platycladus orientalis, Stipa bungeana, etc.

2, To add a description of soil water sampling time;

**Response: Done.** We have revised it.

Soil samples in the unsaturated zone were collected from May 2013 to October 2015. Soil was sampled at 10 cm intervals for the first 40 cm, 20 cm intervals from 40 to 100 cm, 30 cm intervals from 100 to 130 cm. Maximum depths of sampling ranged up to 130 cm (Plant root is rarely found below 100 cm in the study area). At each sampling site, soil moisture (volumetric soil water content) was obtained with time domain reflectometry (TDR) in the field manually at 0-10, 10-20, 20-30, 30-40, 40-60, 60-80, 80-100and 100-130 cm. Soil moisture content was determined by oven drying method simultaneously.

3, Keywords should be hydrogen and oxygen stable isotope;

**Response: Done.** We have revised it.

4, How many precipitation samples were there to define LMWL? Please add sample number.

**Response: Done.** 34 precipitation samples were collected from May 2013 to October 2015.

5, To increase the new relevant references.

**Response: Done.** We have added new relevant references

---

## Author Comment (AC5) · 1 Feb 2017

Soil moisture for vegetation growth is very important in Loess Plateau. Soil water migration is essential to describe the movement of salt, carbon, nitrogen and other nutrient. Although there are many studies on the soil moisture, there are less works on the mechanism of soil water migration in the unsaturated zone in Loess Plateau of arid region in China based on successive observations and isotopes. The scientific issue of this paper is sure, research idea is clear, research method is correct, and research results are reliable. From my point of view, the work is well-done and provides reliable results to soil water migration in arid region of China.

1. Adding relevant detailed introduction about soil and hydrology in the research area.

**Response: Done.** We have revised it. The same as above.

The watershed area is $8.91km^2$, which belongs to the hilly and gully region of Loess Plateau. Gully density is $3.14km/km^2$, and the ditch depth ranges from 30 to 50m. Soil type is yellow loessal soil and saline soil, and the average thickness ranges from 40 to 60m. The soil density of soil layer ranges from 1.1 to 1.4 $g/cm^3$, average soil porosity is 55%. Soil structure has a vertical joint, and the nature of soil is loose and its wet collapsibility is serious. The grassland and shrubland ecosystems are the most extensively dominant ecosystems in the Anjiugou river basin. As it is a representative area of Loess Plateau area, the Anjiagou River basin is a suitable area for soil water study in semi-arid region.

The study area has broken terrain and serious soil erosion, with the terrain being loess long beam and terraces, and gully valley landscape. Geological structure is the uplift zone between the eastern part of Qilian fold system and the west Qinling fold system, at an altitude of 1700 m ~ 2580 m, with the gully density being 3 ~ 5 $km/km^2$, ditch slope being 5 ~ 10%, and the mountain slope being generally 20 to 50. The sunny slope is steep, while the shade slope is relatively flat.

The soil parent material is quaternary eolian loess, and the zonal soil mainly is yellow spongy soils, sierozem, which belongs to the typical semiarid loess hilly-gully region. It has soft soil, homogeneous structure, thicker soil layer, good water performance, and the widest distribution. The average thickness is 40 ~ 60 m. Clay soil is between 33.12% ~ 42.17%, organic matter content is between 0.37% ~ 1.34%, soil bulk density is 1.17 $g/cm^3$, wilting moisture content is 7.3%, and the saturated moisture content is 21.9% at the 0 ~ 20 cm. The soil bulk density is 1.09~1.36g $cm^{-3}$, and the porosity is 50% ~ 55% at the 2 m soil layer. The soil has vertical joint and strong collapsibility, so the

soil erosion is easily happened, and the soil erosion modulus is 5000 t/km$^2 \cdot$ a.

2. As mentioned in the paper, "Soil moisture content was determined by oven drying method simultaneously" How much about the depth of soil sample? Does every depth be concluded respectively? Does the measured depth be in accordance with that of time domain reflectometry ?

   **Response: Done.**

   Soil samples in the unsaturated zone were collected at 10 cm intervals from 100 to 130 cm. Maximum depths of sampling ranged up to 130 cm, with every depth at 0-10, 10-20, 20-30, 30-40, 40-60, 60-80, 80-100and 100-130 cm being determined by oven drying method. The measured depth were accordance with that of time domain reflectometry (TDR).

3. Please explain the reason for choosing caraganakorshinshiikom as the object of study, and illustrate its characteristics in the part of research area summary.

   **Response: Done.** We have revised it.

   The grassland and shrubland ecosystems were the most extensive dominant ecosystems in Loess Plateau. The area accounts for about 50% of the total area, and it is the main part of loess plateau surface cover. Caraganakorshinshiikom is the dominant species in the study area. It is most widely distributed. Therefore, we choose caraganakorshinshiikom as the object of the study.

4. There are many mentioned in the paper, such as "Plant Xylem" "The isotopic composition of soil layer is similar to the xylem water is confirmed by comparing _D and _18O of xylem water and soil water, which reflect the signatures of soil at the depth of soil water uptake by plants." "soil water from the surface horizons (20-40cm) contributed to 8%-21% of plant xylem water," plant roots. Is it the xylogen and root system of a certain kind of plant? Caraganakorshinshiikom?

   In general, this is a well-written paper containing reliable results which merits publication.

   **Response: Done.** The xylogen and root system is Caraganakorshinshiikom,

---

## Author Comment (AC6) · 1 Feb 2017

The authors have tried to do some fundamental research based on field survey and water samples analyses. The study sampled soil profile water stable isotopes of soil, and discussed soil water-plant water recharge sources. This study provides some useful information for soil moisture dynamics and stable characteristics. So it is significant.

However, I could recommend the paper for publication after the following issues have been addressed by the authors:

P1 L19, D is used in the text for the first time, so please state this is Deuterium.

**Response: Done.** We have revised it. It should be deuterium when D is used in the text for the first time.

P4 , L 80ã˘AA¸ "groundwater in the unsaturated zone" There is no relevant analysis about

in the whole passage, so it should be deleted. P7 , L52-53ã˘AA¸ P8 ,167

"The measured D and18 O of soil water range from -72.42‰ to -37.05‰ and -11.74‰

to -3.57‰ respectively." Add the little "delta" before 18O such as _ 18O.

  **Response: Done.** We have revised it. We have deleted the groundwater and added the little "delta" before $^{18}$O.

p10, L 230-233ïijN˘ " Most of the sampling sites in Anjiagou River basin (except 14-7-20 and 14-9-24 sites), the isotopic composition of soil water in shallow layer are enriched in D and $^{18}$O due to evaporation. The isotopic composition of soil water with depth in the soil profile. " Here the changes should be specifically explained.

**Response: Done.** Precipitation is infiltrating and completely mixing with free water. The vertical trend in δD and δ$^{18}$O profile of soil water is simple. There is an abrupt peak value in the isotopic profiles,

which suggests that the older water is pushed downward by the new water and infiltrates to the deep soil.

p12, L 261-264ïijŇ "the preferred flow is rarely found in the Loess Plateau, except that there are which is caused by plant root system or animal invasion, etc." "The macrospores can be found in the 100-200cm soil layer in the study area, which are caused by plant root or animal activity." Here "plant root or animal activity" should be changed into "plant root or animal invasion", and exact illustration is needed.

**Response: Done.** We have revised it. The macrospores are caused by plant root in Loess Plateau.

P13 , L 292ã˘AA¸ "often-used" Here "often-used" should be changed into "often used".

**Response: Done.** We have revised it into "often used".

---

## Author Comment (AC7) · 1 Feb 2017

The literature search is not current enough and misses some important recent papers:

L44: This is completely wrong. Read and include the references of Gaj et al., 2017; Oerter et al., 2014, Meibner et al., 2014; Lin and Horita, 2016.

L45-46: Need to include references to support this claim. I suggest the authors read and include citations from Sprenger et al. 2016.

L44-46,"The isotopes of D and   18 O are conservative and do not react with clay minerals and other soil materials. D and 18 O are widelyused to investigate ecological and hydrological processes. Stable isotopes can provide information about mixing, transport   processes and residence time of water within a soil profile in the unsaturated zone."

**Response: Done. We have added some current papers.**

"The isotopes of D and   18 O are conservative and do not react with clay minerals and other soil materials. D and 18 O are widely used to investigate ecological and hydrological processes. Stable isotopes can provide information about mixing, transport processes and residence time of water within a soil profile in the unsaturated zone."

At present, there are already many relevant research findings, as follows:

Carey Gazis , Xiahong Feng ,A stable isotope study of soil water: evidence for mixing and preferential flow paths,Geoderma 119 (2004) 97–111

Barnes, C. J. and Allison, G. B.: Tracing of water movement in the unsaturated zone using stable isotopes of hydrogen and oxygen, J. Hydrol., 100, 143-176, 1988.

Arny, E., Svein, B. D., Hans, C., Steen, L., Thorsteinn, J., Sigfus, J. and Johnsen,: Monitoring the water vapor isotopic composition in the temperate North Atlantic, J. Geop. Res., Abstracts, 15, 2013-5376, 2013.

Busari, M. A., Salako, F. K., Tuniz, C. and Zuppi, G. M.: Estimation of soil water evaporative loss after tillage operation using the stable isotope technique, Int. Agrophys., 27, 257-264, 2013.

Gazis, C. and Feng, X. H.: A stable isotope study of soil water: evidence for mixing and preferential flow paths, Geoderma, 119, 97-111, 2004.

Hooper, R. P.: Diagnostic tools for mixing models of stream water chemistry, Water Resour. Res.,39,

1055, 2003.

Komor, S. C. and Emerson, D. G.: Movements of water, solutes, and stable isotopes in the unsaturated zones of two sand plains in the upper Midwest, Water Resour. Res., 30, 253-267, 1994.

Parnell, A. C., Inger, R., Bearhop, S. and Jackson, A. L.: Source Partitioning Using Stable Isotopes: Coping with Too Much Variation, Plos. One, 5, 9672, 2010.

Phillips, S. J., Anderson, R. P. and Schapire, R. E.: Maximum entropy modeling of species geographic distribution, Ecol. Model, 19, 231-259, 2013.

*Busari M.A., Salako F.K., Tuniz C., Zuppi G.M.* Estimation of soil water evaporative loss after tillage operation using the stable isotope technique, *Int. Agrophys., 2013, 27, 257-264*

Shen Y. J., Z. B. Zhang, L. Gao, X. Peng. Evaluating contribution of soil water to paddy rice by stable isotopes of hydrogen and oxygen. Paddy Water Environ,DOI 10.1007/s10333-013-0414-y

Pei Zhao, Xiangyu Tang, Peng Zhao,Identifying the water source for subsurface flow with deuterium and oxygen-18 isotopes of soil water collected from tension lysimeters and cores, Journal of Hydrology 503 (2013) 1-10.

Donald L. Phillips.Mixing models in analyses of diet using multiple stable isotopes: a critique, Oecologia (2001) 127:166–170

Rodney A. C, David J. C. Using stable oxygen isotopes to quantify the water source used for transpiration by native shrubs in the San Luis Valley, Colorado U.S.A. *Plant and Soil* 260: 225–236, 2004.

L115: This needs some justification. There must be some discussion (somewhere in the paper, but probably not in the methods section) of the many papers that cite problems with the vacuum extraction process and variability in its results, see

Araguas et al., 1995; Orlowski et al., 2016a; Orlowski et al, 2016b. Details on the vacuum distillation methods need to be included.

References that need to be included:

Araguas-Araguas, L., Rozanski, K., Gonfiantini, R., Louvat, D., 1995. Isotope effects accompanying

vacuum extraction of soil water for stable isotope analyses. J Hydrol,

168: 159-171.

Gaj, M. et al., 2017. Mineral mediated isotope fractionation of soil water. Rapid Communications in

Mass Spectrometry, 31(3): 269-280.

Lin, Y., Horita, J., 2016. An experimental study on isotope fractionation in a mesoporous silica-water

system with implications for vadose-zone hydrology. Geochimica et Cosmochimica Acta, 184:

257-271.

Meißner, M., Köhler, M., Schwendenmann, L., Hölscher, D., Dyckmans, J., 2014. Soil water uptake by

trees using water stable isotopes (2H and 18O) a method test regarding soil moisture, texture and

carbonate. Plant and Soil, 376(1-2): 327-335.

Oerter, E. et al., 2014. Oxygen isotope fractionation effects in soil water via interaction with cations

(Mg, Ca, K, Na) adsorbed to phyllosilicate clay minerals. J Hydrol, 515:1-9.

Orlowski, N., Breuer, L., McDonnell, J.J., 2016a. Critical issues with cryogenic extraction of soil water

for stable isotope analysis. Ecohydrology, 9: 3-10.

Orlowski, N., Pratt, D.L., McDonnell, J.J., 2016b. Intercomparison of soil pore water extraction

methods for stable isotope analysis. Hydrol Process, 30(19): 3434-3449.

Sprenger, M., Leistert, H., Gimbel, K., Weiler, M., 2016. Illuminating hydrological processes at the

soil - vegetation - atmosphere interface with water stable isotopes. Reviews of Geophysics.

**Response:** Water was extracted from soil, leaf, branch, xylem and root by cryogenic vacuum distillation method. The moisture in the soil or plants under the condition of vacuum (vacuum below60MT), was heated by heat set to 105 ℃ after evaporation. Water vapor of evaporation in -50 ℃ (liquid nitrogen) was collected with frozen water collecting pipe (top-down frozen, in order to increase the collecting rate), and the precision extract of δD and $\delta^{18}$ are ±3‰ and ±0.3‰, respectively.

The vacuum extraction has been well employed in many research works, as follows:

Yonggang Yang, Honglang Xiao , Zuodong Qin,Songbing Zou, et al. Hydrogen and oxygen isotopic

  records in monthly scales variations of hydrological characteristics in the different landscape zones

  [J].Journal of Hydrology, 2013,499:124~131.

Yonggang Yang, Honglang Xiao , Songbing Zou, et al. Hydrological processes in the different

landscape zones of alpine cold regions in the wet season, combining isotopic and hydrochemical tracers[J]. Hydrological Processes, 2012, 25: 1457~1466.

Yonggang Yang, Honglang Xiao , Songbing Zou, et al. Hydrologic processes in the different landscape zones in the alpine cold region during the melting period [J]. Journal of Hydrology, 2011, 409:149~156.

West AG, Patrickson SJ, Ehleringer JR.Water extrac-tion times for plant and soil materials used in stable isotope analysis. Rapid Commun Mass Spectrom (2006)20: 1317–1321

EhleringerJR, OsmondCB. Stableisotopes.In:PearcyRW, Ehleringer JR, Mooney HA, Rundel PW (eds) Plant physio- logical ecology: Field methods and instrumentation.Chapman and Hall, New York, (1989).pp 281–300

Carey Gazis , Xiahong Feng ,A stable isotope study of soil water: evidence for mixing and preferential flow paths,Geoderma 119 (2004) 97–111

Rodney A. C ,  David J. C, Using stable oxygen isotopes to quantify the water source used for transpiration by native shrubs in the San Luis Valley, Colorado U.S.A. Plant and Soil. 2004.260: 225–236,

Meik Meißner , Michael Köhler , Luitgard Schwendenmann,Dirk Hölscher , Jens Dyckmans, Soil water uptake by trees using water stable isotopes ($\delta$2Hand $\delta$18O)−a method test regarding soil moisture, texture and carbonate Plant Soil .2014. 376:327–335

Stumpp. C. , Maloszewski,Stichler P W. , Fank. J.Environmental isotope (d18O) and hydrological data to assess water flow in unsaturated soils planted with different crops: Case study lysimeter station"Wagna" , Journal of Hydrology 369 .2009:198–208

Xianfang Song, Shiqin Wang1, Guoqiang Xiao, A study of soil water movement combining soil water potential with stable isotopes at two sites of shallow groundwater areas in the North China Plain Hydrol. Process. 2009.23, 1376–1388.

---

## Author Comment (AC8) · 1 Feb 2017

The comment was uploaded in the form of a supplement:
http://www.hydrol-earth-syst-sci-discuss.net/hess-2016-555/hess-2016-555-AC8-supplement.pdf

---

## Author Comment (AC9) · 1 Feb 2017

The comment was uploaded in the form of a supplement:
http://www.hydrol-earth-syst-sci-discuss.net/hess-2016-555/hess-2016-555-AC9-supplement.pdf
* * *